# Food inflation and child undernutrition in low and middle income countries

Derek Headey [1] ✉ & Marie Ruel [2]

The 21st Century has been marked by increased volatility in food prices, with global price spikes in 2007-08, 2010-11, and again in 2021-22. The impact of food inflation on the risk of child undernutrition is not well understood, however. This study explores the potential impacts of food inflation on wasting and stunting among 1.27 million pre-school children from 44 developing countries. On average, a 5 percent increase in the real price of food increases the risk of wasting by 9 percent and severe wasting by 14 percent. These risks apply to young infants, suggesting a prenatal pathway, as well as to older children who typically experience a deterioration in diet quality in the wake of food inflation. Male children and children from poor and rural landless households are more severely impacted. Food inflation during pregnancy and the first year after birth also increases the risk of stunting for children 2-5 years of age. This evidence provides a strong rationale for interventions to prevent food inflation and mitigate its impacts on vulnerable children and their mothers.

Food prices have become extremely volatile in the 20th Century. After decades of secular decline, international cereal prices rose dramatically in the mid-2000s, spiking in 2007–08 in what was widely termed a global food crisis[1] (Fig. 1). Prices plummeted briefly during the 2009 global financial crisis but spiked again in 2010–11 before gradually declining over several years. However, COVID-19 tailwinds in the form of both supply-side disruptions and macroeconomic distortions led to rapid food inflation in 2021, which was then exacerbated by the conflict between Russia and Ukraine—two of the biggest food exporters in the world—in early 2022[2]. By March 2022 the FAO[3] food price index had reached an all-time high, 116% greater than its 2000 value.

Such dramatic increases in international food prices are clearly a major threat to the world's poorest and most malnourished populations, but research on the impacts of food inflation on undernutrition in low and middle income countries (LMICs) is scarce. Studies of food prices and household poverty suggest that the impacts of higher food prices on economically disadvantaged households are heterogenous across livelihoods and over time. The Nobel prize-winning economist, Angus Deaton, argued that the short-run impact of higher food prices on a household's income—an important predictor of child nutritional status[4] and diet quality—depends on whether a household is a net food

consumer or a net food producer[5]. Analyses using Deaton's method find that most poor households are net food consumers so generally suffer losses in income as food prices increase[6–14]. However, Deaton's method deliberately omits medium term adjustments for household's use of adaptive coping mechanisms (such as substitutions in foods consumed or produced) and economywide adjustments (such as increases in rural wages emanating from increased farmer demand for hired labor). Medium-run empirical analyses that allow for these adaptations have found that higher food prices tend to reduce poverty, at least in more rural economies[15–22].

The impact of higher food prices on undernutrition among young children is conceptually even more complex. Child wasting—low weight-for-height—is sensitive to short-term shocks such as declines in diet quality/quantity or infections and other illnesses. For this reason, wasting, a measure of acute malnutrition, is widely used for nutritional surveillance in shock-prone settings because it is an early marker of nutritional deterioration that requires immediate attention given its strong association with early childhood mortality[23]. In contrast, child stunting or short stature (low height-for-age) captures the cumulative or longer-term impacts of poor nutrition (including repeated episodes of wasting)[24], especially during the period spanning from conception

[1]Senior Research Fellow, The International Food Policy Research Institute (IFPRI), Colombo, Sri Lanka. [2]Senior Research Fellow, The International Food Policy Research Institute (IFPRI), Washington, DC, USA. ✉e-mail: d.headey@cgiar.org

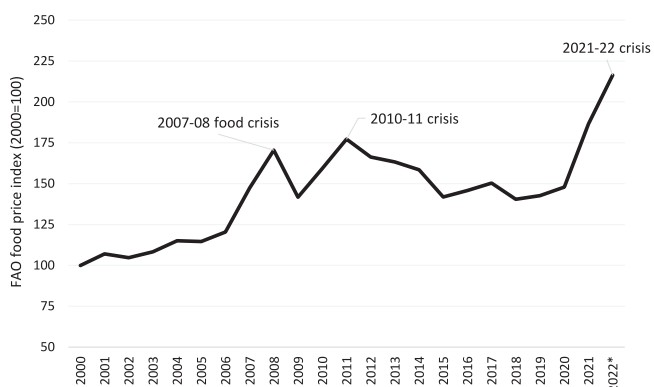

**Fig. 1 | Trends in the FAO real international cereal price index from 2000 to August 2022.** Notes: Source data are provided as a Source data file. Data are the author's construction from the real FAO cereal price index[3]. The 2022 value refers to the average of January-August 2022. The index is based so that 2000 = 100 and consists of the average values of 5 commodity group price indices (cereals, meat, vegetables, edible oils, and sugar). Each sub-index is a weighted average of a range of export price quotations. See the FAO's food price webpage for further details: https://www.fao.org/worldfoodsituation/foodpricesindex/en/.

to approximately two years of age, referred to as the first 1000 days of life[25]. A number of studies link climate shocks to wasting or stunting among pre-schoolers[26–28], or even short stature in adulthood[29,30], while other studies examine the impacts of macroeconomic crises on these nutrition outcomes[31]. Such crises often involve food inflation, but are also characterized by rising unemployment and declines in nominal incomes, making it difficult to isolate the specific impacts of food inflation on nutrition outcomes.

Two studies that we are aware of at least try to examine the impact of food price changes on undernutrition in isolation from broader macroeconomic crises. One study used long-term but high-frequency nutrition surveillance data from Bangladesh to descriptively show that when rice prices increase, households spend less on non-staple foods and child underweight (low weight-for-age) prevalence increases[32]. More related to the present study is an analysis of the 2008–09 food price crisis using a propitiously timed household survey in Mozambique that covered both low and high food inflation periods[33]. Controlling for confounding factors, the study found that children exposed to high inflation were significantly more likely to be wasted and underweight. In summary, despite longstanding speculation that food inflation increases the risk of child or maternal undernutrition[34], only one study from Mozambique has rigorously identified a potential impact of food inflation on wasting risks. The external validity of that finding remains uncertain.

In the present study, we test whether increases in the real price of food (hereafter "food inflation" for short) are a short-term risk factor for child wasting and a longer-term risk factor for child stunting, using a novel database linking 130 Demographic Health Surveys (DHS) implemented in 44 LMICs over 2000–2021 with national-level monthly data on real food price changes. We also assess whether child age at the time of food price shocks affects the magnitude of impacts on wasting or stunting. In addition, we use select demographic and socioeconomic indicators in the DHS to test whether the wasting and stunting risks associated with food inflation vary by rural/urban location, gender, poverty, and farm ownership status, with potential implications for targeting of scarce programming resources during national or international food crises. We also test whether food inflation leads to a deterioration in child diet quality or increases in symptoms of infection, as plausible pathways of impact.

The remainder of this study is structured as follows. The "Results" section describes our results, focusing first on our key results and then extensions and robustness tests. The "Discussion" section the

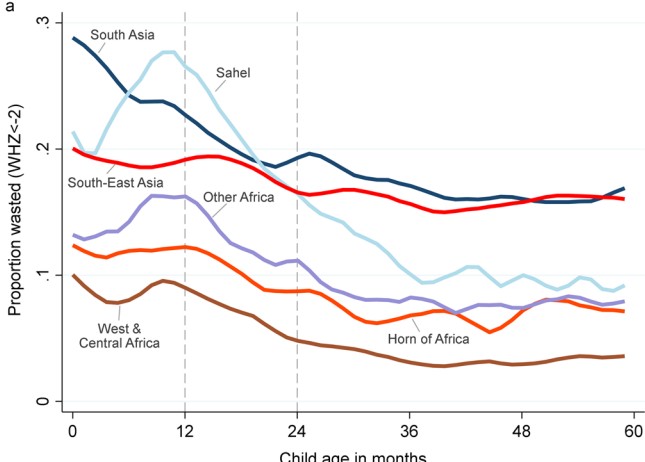

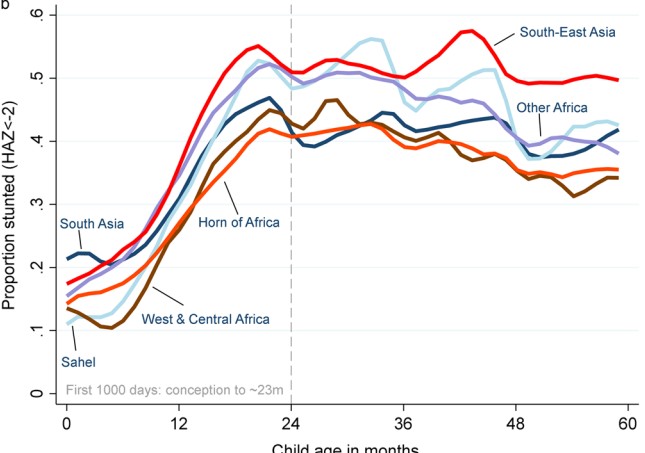

**Fig. 2 | Local polynomial regression estimates of wasting and stunting prevalence by child age for DHS regions with high undernutrition burdens.** Notes: **a** Wasting (WHZ < −2) by age and region. **b** Stunting (HAZ < −2) by age and region Authors' estimates from DHS data using the lpoly command in STATA™. See the "Methods" section for more details.

implications of these findings for policies, programs, and research. The "Methods" section details our data and methods in detail.

## Results

Our results are structured in four parts: (1) descriptive results designed to outline patterns in wasting, stunting, and food inflation; (2) regression results for child wasting and stunting; and (3) sensitivity tests and extensions.

### Descriptive results

Our sample of 1.271 million children 0–59 months of age in 44 LMICs shows large variation in wasting and stunting prevalence across countries. Approximately 13% of the sample suffers from wasting (<−2 WHZ) while 5% are severely wasted (<−3WHZ) (Supplementary Table 1). Stunting is much more common, with 35% of the sample stunted (HAZ < −2) and 15% severely stunted (HAZ < −3). Wasting and stunting prevalence vary by region and by child age, with important implications for regression analyses[35].

Figure 2 reports smoothed regression estimates of wasting and stunting by child age for DHS sub-regions with the highest child undernutrition burdens, all of which are in sub-Saharan Africa and Asia. In Panel A we observe large variations in wasting prevalence across regions that seem only weakly associated with overall regional economic development levels. For example, wasting prevalence is highest

in South Asia and South-East Asia, in spite their higher economic development than African regions. The only exception is the high levels of wasting observed in the Sahel region, especially in the first 2 years of life. Wasting patterns also show interesting variations by child age, with almost 30% of South Asian children in our sample being born wasted, compared to ~20% in South-East Asia and in the Sahel and between 10 and 13% in other African regions. In the two Asian regions, wasting prevalence declines gradually from birth to 24 months of age when it reaches around 18–20% and stabilizes thereafter. For African regions (except the Sahel), similar declines are observed over time, especially after 12 months of age and wasting remains much less prevalent than in Asia throughout the full first 60 months of life. The Sahel shows quite different age-related patterns of wasting, with rapid increases during the first year (up to 28% by 10 months of age) and then falls sharply to around 10% by 36 months of age before levelling off like in other regions. Our results showing the highest burden of wasting between birth and 3–6 months are consistent with global evidence[36] and are of great concern given the association between wasting and excess mortality risk in this age group[23].

Stunting, which results from the cumulative effect of repeated or chronic cumulative nutritional insults, shows very different patterns (Panel B, Fig. 2). First, stunting prevalence does not show as much variation across high-burden regions as wasting does. In all Asian and African regions included in our sample, between 10 and 20% of children are born stunted, but stunting prevalence increases gradually from around 4 to 20 months of age, before levelling off thereafter. The process of becoming stunted therefore mostly occurs during the first 1000 days of life, from conception to approximately age 2 years.

Supplementary Fig. 1 uses the same regression smoothing technique (with 95% confidence intervals) to demonstrate heterogeneity in wasting prevalence by demographic, geographic, and socioeconomic characteristics. Male and female children are born with similar wasting risks (Panel A), but wasting is significantly higher for boys compared to girls, which is likely related to the male fragility hypothesis[37]. Panel B shows that wasting is much higher among rural children from birth all the way to age 5 years, with the difference varying between 3–6 percentage points, reflecting the many disadvantages that rural populations have in socioeconomic status and access to nutrition-relevant services, such as healthcare[38,39]. Panel C focuses more explicitly on differences in socioeconomic status and shows higher prevalence of wasting among children from asset-poor households compared to those from non-poor households (see also Supplementary Fig. 2 for asset-poverty comparisons in India specifically). Finally, Panel D focuses on a key measure of rural wealth as well as potential resilience to food price shocks and confirms that children from rural-farm households are less likely to be wasted, typically by around 3 percentage points.

This study exploits variation in the real price of food within DHS countries, as measured by changes in the ratio of the consumer price index (CPI) for food to the CPI for all consumer items. We interchangeably refer to increases in this food CPI/total CPI ratio as either food inflation or real food price increases. For wasting regressions, we measure real food price changes for the three months prior to the month of anthropometric measurement[40], but for stunting and child diet diversity regressions we use 12-month food price changes in the prenatal period and first and second years of life. In all our regression analyses we essentially exploit the fact that DHS rounds are conducted in both low food inflation and high food inflation periods, including a number of surveys conducted in the 2007–2011 period when international and domestic food prices were highly volatile. Figure 3 demonstrates this by reporting mean, minimum, and maximum 12-month food price changes in the 130-round DHS sample. There are large temporal and cross-country variations in real food price changes through most of the period in question. There were also 38 DHS surveys conducted between 2007–2011 when international and domestic

price volatility was very high, with striking instances of 3-month food inflation in Liberia (10% in April 2007), Bolivia (15% in June 2008), Kenya (18% in January 2009), Ethiopia (13.6% in mid-2011) and Uganda (20% in mid-2011). Supplementary Fig. 3 also reports a histogram of the distribution of 3-month real food inflation measure used for the wasting analysis.

## Main regression results: food inflation and child wasting and stunting

Figure 4 reports coefficient plots from weighted regressions that represent the predicted impact of 5% increases in the food/total CPI ratio for the past three months on wasting (black circles) and severe wasting risks (blue circles), with 95% confidence intervals. The regression approach follows a previous study on macroeconomic

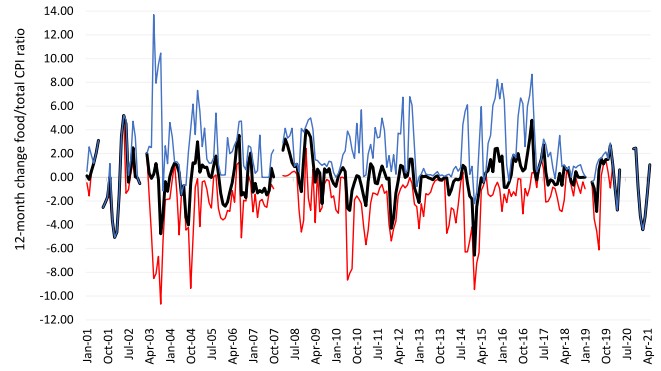

**Fig. 3 | Mean, minimum, and maximum food inflation in the 12 months prior to anthropometric measurement in the 130 DHS rounds.** Notes: Source data are provided as a Source data file. Authors' calculations from FAO consumer price database for the DHS surveys in our regression sample[40]. See the "Methods" section for more details.

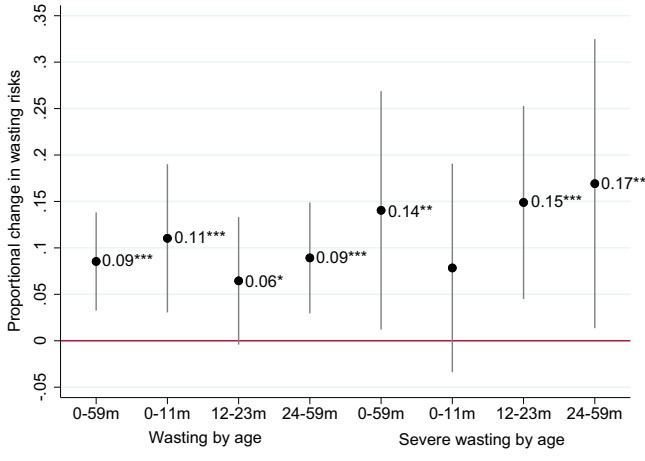

**Fig. 4 | Weighted linear probability coefficients representing the impact of a 5% increase in the real food price index over the 3 months prior to wasting measurement, stratified by child age.** Notes: 95% confidence intervals based on standard errors clustered at the country level are reported in parentheses. *, **, and *** represent statistical significance at the 10%, 5%, and 1% level, respectively, from a two-sided test. The regressions control for DHS-based predictors of wasting, country fixed effects, and region-specific time trends, seasonality factors, and wasting-age dynamics, and are weighted to be representative of this specific sample of DHS countries. Changes in the food/total CPI ratio are interacted with a country's mean wasting prevalence across all its survey rounds to ensure that the impact of inflation is proportional to a country's typical wasting prevalence. See the "Methods" section for more details. The full sample includes 1.271 children in 44 LMICs.

shocks and child wasting[41] in controlling for DHS-based predictors of wasting, country fixed effects, and region-specific time trends, seasonality factors, and wasting-age dynamics. Regressions are weighted to be representative of this specific sample of DHS countries, and changes in the food/total CPI ratio are interacted with a country's mean wasting prevalence across all its survey rounds to ensure that the impact of inflation is proportional to a country's typical wasting prevalence. This is important because countries with very low and very high levels of wasting are unlikely to see the same absolute change in wasting risks from a given macroeconomic shock[41]; for example, countries with low wasting rates will have very few children close to the −3 z-score threshold for severe wasting, so even a large food price shock will have little impact on severe wasting prevalence in an absolute sense.

Due to the striking wasting-age patterns described above, the results are stratified by child age. In the full sample of children 0–59 months the regression coefficient is positive and highly statistically significant, implying that a 5% increase in the food/total CPI ratio—equivalent to around two standard deviations in this sample—predicts a 9% increase in the risk of wasting. This marginal effect varies from 11% for children 0–11 months of age to 6% for children 12–23 months of age. Coefficients for severe wasting are somewhat larger in magnitude but less precisely estimated. A 5% increase in real food prices predicts a 14% increase in severe wasting for children 0–59 months, with similar magnitudes of effects in the 11–23 and 24–59 month samples. The coefficient for the 0–11-month age group is not statistically significant in a two-sided test.

Given the large and statistically significant elasticity of wasting with respect to food inflation in the 0–11 month sample in Fig. 4, it seems plausible that food inflation increases the risk of low birthweight (or wasting at birth) by adversely affecting maternal nutrition during pregnancy. In Fig. 5 we therefore examine wasting risks for infants 0–5 months of age. We find that for both wasting and severe wasting, the coefficients are large and statistically significant, supporting the

hypothesis that a prenatal maternal nutrition mechanism links food inflation to wasting at birth and in the first few months of life. This result is important because mortality rates among newborns and young infants are especially high, suggesting food inflation poses a major risk for infant mortality through a maternal malnutrition pathway.

## Testing for heterogenous effects of food inflation on wasting by sociodemographic groups

Biological and economic theories suggest that the impacts of food inflation on undernutrition may be heterogeneous in other dimensions, and Supplementary Fig. 1 demonstrated significant disparities in wasting prevalence across demographic and socioeconomic strata. Table 1 therefore introduces interaction terms between food inflation and these strata for moderate/severe wasting, while Supplementary Table 5 reports results for severe wasting.

In regression (1) we observe that food inflation's wasting risks are around twice as large for boys as they are for girls. In regression (2) we find that children from urban households are less likely to become wasted after food inflation compared to their rural counterparts. While surprising—urban households are mostly net-food consumers highly dependent on markets—it is also true that urban children are less likely to have low WHZ scores to begin with, and also have better access to health and nutrition services, and more assets and parental education. In regression (3) we find that asset-poor children are much more likely to become wasted than non-poor children. A 5% increase in real food prices increased the risk of wasting by just 6% for non-poor children, but by 15% for asset-poor children.

In regression (4) we turn to a rural sample to test whether food inflation's impact on wasting depends on whether a household owns land (and can therefore produce food) or is landless (and therefore dependent on market purchases). The coefficient on farmland ownership is highly significant and reduces the impacts of food inflation on wasting by nearly half. In regression (5) we focus on the full sample again and test multiple interactions (except farmland ownership) and find that the interactions persist in the presence of each other, although the partial protection of living in an urban area is reduced when the poverty interaction is added. In regression (6) we focus on the rural sample and include all the interactions, including farmland ownership. All the interaction coefficients remain statistically significant, although the magnitude of the coefficient on poverty actually increases in absolute magnitude.

The presence of multiple statistically significant coefficients in regressions (5) and (6) implies additive effects. For example, the worst affected group are children from landless rural households that are also asset poor: regression (6) predicts that a 5% increase in the real price of food for this highly vulnerable group results in a 29 percent increase in the risk of wasting for boys and a 23% increase in the risk of wasting for girls. We also note that results for severe wasting are qualitatively very similar to those reported in Table 1 (See Supplementary Table 5).

## Food inflation in the first 1000 days of life and subsequent stunting risks

Even relatively brief nutritional insults can have longer-term consequences on child growth and development, especially insults that occur in the first 1000 days of life when children are highly vulnerable stunting rates climb precipitously in LMICs (Fig. 2, Panel B). We therefore tested whether food inflation in the prenatal period or the first or second years after birth is a longer-term risk factor for stunting in the 24–59 month period (after the first 1000 days). Figure 6 reports results from separate regressions for stunting and severe stunting. For both indicators we find that food inflation in the prenatal period or the first year after birth significantly elevates the risk of stunting in the 24–59 month period, while food inflation in the second year of life also

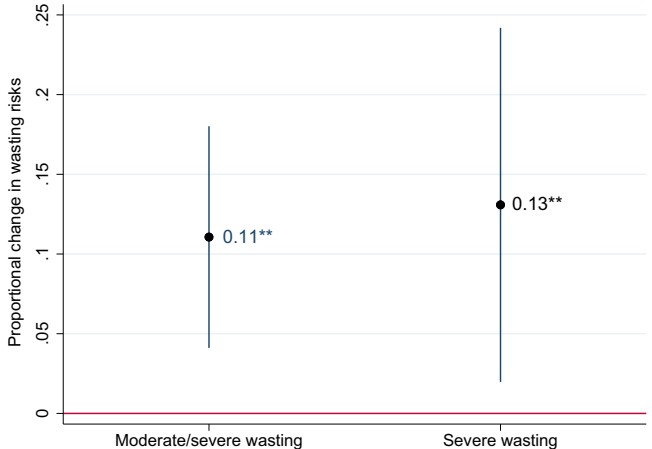

**Fig. 5 | Weighted linear probability coefficients representing the impact of a 5% increase in the real food price index in the past 3 months on wasting and severe wasting among children 0–5 months of age.** Notes: 95% confidence intervals based on standard errors clustered at the country level are reported in parentheses. *, **, and *** represent statistical significance at the 10%, 5%, and 1% level, respectively, from a two-sided test. The regressions control for DHS-based predictors of wasting, country fixed effects, and region-specific time trends, seasonality factors, and wasting-age dynamics, and are weighted to be representative of this specific sample of DHS countries. Changes in the food/total CPI ratio are interacted with a country's mean wasting prevalence across all its survey rounds to ensure that the impact of inflation is proportional to a country's typical wasting prevalence. See the "Methods" section for more details. The full sample includes 1.271 children in 44 LMICs.

**Table 1 | Weighted multivariate linear probability models of wasting risks as a function of 5% increases in the real food price index over the past 3 months interacted with urban locality, gender, asset poverty, and farmland ownership**

| Interaction variable Sample | (1) Girl child Full | (2) Urban location Full | (3) Asset-poverty Full | (4) Owning farmland Rural | (5) Multiple Full | (6) Multiple Rural |
|---|---|---|---|---|---|---|
| Food inflation (base group) | 0.11 | 0.10 | 0.06 | 0.16 | 0.10 | 0.16 |
| | $p = 0.00$ | $p = 0.00$ | $p = 0.01$ | $p = 0.00$ | $p = 0.00$ | $p = 0.00$ |
| | (0.06, 0.16) | (0.05, 0.16) | (0.02, 0.11) | (0.13, 0.20) | (0.06, 0.15) | (0.12, 0.19) |
| Food inflation*girl | −0.05 | | | | −0.05 | −0.06 |
| | $p = 0.00$ | | | | $p = 0.00$ | $p = 0.00$ |
| | (−0.07, −0.03) | | | | (−0.07, −0.03) | (−0.07, −0.05) |
| Food inflation*urban | | −0.06 | | | −0.04 | |
| | | $p = 0.00$ | | | $p = 0.01$ | |
| | | (−0.10, −0.02) | | | (−0.07, −0.01) | |
| Food inflation*asset-poor | | | 0.09 | | 0.08 | 0.13 |
| | | | $p = 0.01$ | | $p = 0.03$ | $p = 0.00$ |
| | | | (0.02, 0.16) | | (0.01, 0.14) | (0.08, 0.18) |
| Food inflation*farmland | | | | −0.07 | | −0.07 |
| | | | | $p = 0.00$ | | $p = 0.00$ |
| | | | | (−0.09, −0.04) | | (−0.09, −0.04) |
| Girl child | −0.02 | −0.02 | −0.02 | −0.02 | −0.02 | −0.02 |
| | $p = 0.00$ | $p = 0.00$ | $p = 0.00$ | $p = 0.00$ | $p = 0.00$ | $p = 0.00$ |
| | (−0.02, −0.01) | (−0.02, −0.01) | (−0.02, −0.01) | (−0.02, −0.01) | (−0.02, −0.01) | (−0.02, −0.01) |
| Urban locality | 0.01 | 0.01 | 0.01 | | | |
| | $p = 0.00$ | $p = 0.00$ | $p = 0.00$ | | | |
| | (0.00, 0.01) | (0.00, 0.01) | (0.00, 0.01) | | | |
| Asset-poor | 0.03 | 0.03 | 0.03 | 0.03 | 0.03 | 0.04 |
| | $p = 0.00$ | $p = 0.00$ | $p = 0.00$ | $p = 0.00$ | $p = 0.00$ | $p = 0.00$ |
| | (0.02, 0.03) | (0.02, 0.03) | (0.02, 0.04) | (0.03, 0.04) | (0.02, 0.04) | (0.03, 0.05) |
| Farmland ownership | | | | −0.00 | | −0.00 |
| | | | | $p = 0.58$ | | $p = 0.53$ |
| | | | | (−0.01, 0.00) | | (−0.01, 0.00) |
| Observations | 1,271,886 | 1,271,886 | 1,271,886 | 719,457 | 1,271,886 | 719,457 |
| R-squared | 0.06 | 0.06 | 0.06 | 0.05 | 0.05 | 0.05 |

Notes: Standard errors clustered at the country level are reported in parentheses. *, **, and *** represent statistical significance at the 10%, 5%, and 1% level, respectively, from a two-sided test. The regressions control for DHS-based predictors of wasting, country fixed effects, and region-specific time trends, seasonality factors, and wasting-age dynamics, and are weighted to be representative of this specific sample of DHS countries. Changes in the food/total CPI ratio are interacted with a country's mean wasting prevalence across all its survey rounds to ensure that the impact of inflation is proportional to a country's typical wasting prevalence. See "Methods" for more details. The full sample includes 1.271 children in 44 LMICs.

has positive, but non statistically significant coefficients (in two-sided tests). For moderate/severe stunting the findings suggest that a 5% increase in food prices in the prenatal period increases the risk of stunting by 1.6 percent, and by 1.8 percent in the first year after birth. The point estimates are around twice as large for severe stunting but are still much lower than for wasting. The fact that food inflation in the prenatal period is a strong predictor of later-life stunting is consistent with the indirect evidence reported in Fig. 5 that inflation during pregnancy might affect intra-uterine growth and birthweight (proxied here by wasting during the first few months of life) We also tested for heterogeneous effects of food inflation by introducing the interaction terms used in Table 1, although there is no clear evidence that the same interaction effects hold for stunting (Supplementary Table 7).

**Assessing dietary diversity and symptoms of infections as likely mechanisms linking food inflation to wasting and stunting**
Whilst wasting and stunting are affected by both diets and health, it is likely that the main mechanism linking food inflation to wasting is maternal nutrition and diets during pregnancy and the adequacy of infant and young child feeding practices and diets during postnatal life. The DHS allows measurement of minimum dietary diversity (MDD)

(a proxy for diet quality) for children 6 months and older (although here we reverse the variable to measure inadequate dietary diversity). Adequate diet diversity captures whether a child consumed at least four of seven recommended food groups in the past 24 h. Data on maternal diet diversity is not available in the surveys used, but child and maternal dietary diversity have been shown to be strongly correlated[42]. Child dietary diversity does not capture quantities consumed, but the indicator has been shown to predict mean micronutrient adequacy[43] and energy intake[44]. In our dataset we use regression analysis to show that poor diet diversity predicts an increased risk of wasting by 1.2 percentage points for children 6–23 months, while it increases the risk of stunting by 4.7 points for children 18–23 months (Supplementary Fig. 4). Reported diarrhea and fever symptoms in the previous 2 weeks are also associated with wasting and stunting. For stunting one should probably interpret these associations as indicative of the recent dietary status or illness being reasonable proxies for longer-term dietary status and exposure to disease.

Figure 7 shows that a 5% increase in food inflation in the past 12 months is predictive of a 3% increase in the percentage of children 6–23 months of age with an inadequately diverse diet. This suggests

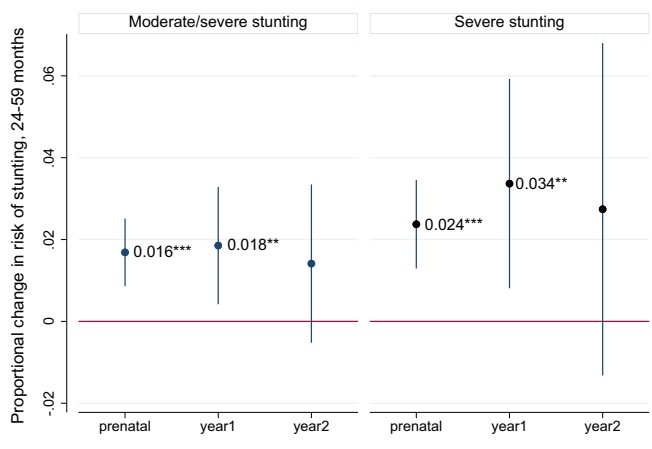

**Fig. 6 | Weighted linear probability coefficients representing the impact of a 5% increase in the real food price index over different periods of the first 1000 days of life on stunting risks among children 24–59 months.** Notes: 95% confidence intervals based on standard errors clustered at the country level are reported in parentheses. *, **, and *** represent statistical significance at the 10%, 5%, and 1% level, respectively, from a two-sided test. The regressions control for DHS-based predictors of stunting, country fixed effects, and region-specific time trends, seasonality factors, and wasting-age dynamics, and are weighted to be representative of this specific sample of DHS countries. Changes in the food/total CPI ratio are interacted with a country's mean stunting prevalence across all its survey rounds to ensure that the impact of inflation is proportional to a country's typical wasting prevalence. See the "Methods" section for more details. The full sample includes 1.271 children in 44 LMICs.

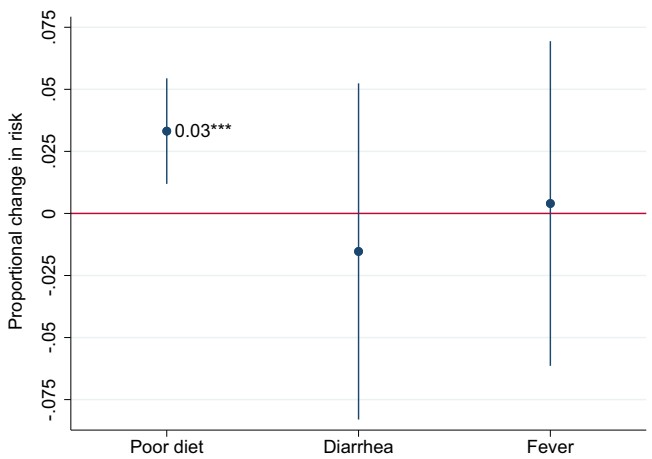

**Fig. 7 | Weighted linear probability coefficients representing the impact of a 5% increase in the real food price index in the past 12 months on risk of children having an inadequately diverse diet, diarrhea, or fever.** Notes: 95% confidence intervals based on standard errors clustered at the country level are reported in parentheses. *, **, and *** represent statistical significance at the 10%, 5%, and 1% level, respectively, from a two-sided test. See the "Methods" section for details on control variables and weighting methods. An inadequately diverse diet (Poor diet) is defined as child not achieving minimum diet diversity, meaning at least four of seven food groups consumed in the past 24 h. Diarrhea and fever refer to any experience of these symptoms in a 2-week recall. The Poor diet regression has a sample size of 300,476 children 6–23 months of age, while the Diarrhea and Fever regressions have samples sizes of 1,257,720 and 1,265,488, respectively.

that deterioration in diet quality is a plausible mechanism by which food inflation affects anthropometric outcomes, at least in children 6–24 months of age. As expected, food inflation was not associated with diarrhea or fever in the past 2 weeks.

## Testing for impacts of total inflation on wasting and stunting

While relative increases in food prices predict increased risks of wasting and stunting, it is possible that non-food inflation could also adversely affect household welfare and child undernutrition. To test this, we added total inflation to the wasting and stunting regression models reported above. The results are presented in Supplementary Figs. 5 (wasting) and Fig. 6 (stunting). Total inflation does not appear to be a statistically significant risk factor for either stunting or wasting. This may be because the poor spend less on non-food goods and services than on food, and because some big-ticket non-food expenditures—such as rent, public healthcare and schooling—are less subject to short-term volatility and inflation, and also have less direct connection to diet quality or disease pathways.

## Discussion

International food prices have become increasingly volatile in recent decades, and food inflation may well be one of the foremost economic challenges of the 21st Century, especially with climatic change and conflict shocks. However, the impacts of food inflation on poverty remain controversial in economics—with seemingly quite different short- and long-term poverty impacts—while the impacts of food inflation on nutrition were largely unknown prior to this study. For 44 LMICs we established that short-term increases in real food prices significantly elevate the risk of child wasting, especially for children from poor and landless households, and for boys more than for girls. Food inflation seems to operate both through shocks to maternal nutrition in pregnancy and postnatal mechanisms. We found strong associations between food inflation and wasting among infants 0–5 months of age, but also that food inflation during the prenatal period predicts stunting later in life (at 24–59 months). However, food inflation also predicts wasting among older age groups, and inadequate dietary diversity among children 6–23 months of age. Given the generally strong association between maternal and child diet diversity[42] and evidence of prenatal inflation affecting birthweight and long-term stunting, it seems likely that an important mechanism linking food inflation and child undernutrition operates through deteriorations in maternal diet quality and nutrition in the wake of rising food prices.

Our results across 44 LMICs are consistent with one previous study rigorously linking wasting and stunting to food inflation in Mozambique[33]. It is also broadly consistent with studies looking at macroeconomic shocks—which often involve food inflation as well other economic impacts—to poor nutrition outcomes[32,45].

These findings come with caveats and limitations. Food prices are measured at the national level only; more granular price data might improve precision, although there is more variation in price shocks across countries than there is within countries. The food CPI index assigns more weight to foods that households typically consume, but future research could examine whether price changes for specific foods influence undernutrition. We also confine our study to wasting and stunting, but it is plausible that inflation shocks also have longer term effects of micronutrient deficiencies[31], and also child mortality. Results for wasting, especially severe wasting among infants, could also be somewhat biased towards zero because wasted children are more likely to die, thus falling out of the sample and leading to a selection bias. To our knowledge, this issue has only previously been explored for stunting[46]. Another limitation comes from the data source, which does not provide information on maternal diet or weight gain during pregnancy and is limited to a dietary diversity proxy indicator in children. Nonetheless, the MDD has been extensively validated in children and found to be a solid proxy for dietary diversity, a key dimension of dietary quality.

There are also limitations in the DHS data. Farm ownership is a useful indicator of rural resilience to food inflation, but data on farm size, irrigation and other aspects of food production would be useful.

There may also be other sources of household resilience to food inflation shocks that we have not identified, such as access to salaried (stable) employment, migration and remittances, social networks, or protective policies and programs. Dietary data for children is also limited to a dietary diversity proxy indicator and a much smaller sample size than other variables. Nonetheless, the MDD has been extensively validated in children and found to be a solid proxy for dietary diversity, a key dimension of dietary quality. The absence of quantitative information on maternal diet or weight gain during pregnancy is another limitation of the DHS data sets, as well as the lack of information on health expenditures, which could be another mechanism linking real income shocks to nutrition outcomes. More research on exactly how food price increases affect nutrition-related mechanisms, especially maternal diets and weight gain during pregnancy, birth outcomes, breastfeeding performance, and children's diets, is certainly warranted.

Bearing these caveats in mind, this study has important implications. Our study suggests that young children are nutritionally highly vulnerable to food price shocks, albeit heterogeneously so. In terms of targeting scarce resources for prevention of undernutrition associated with food inflation, pregnant women and their young infant warrant prioritization, as well as asset-poor households, including the landless poor. Since food inflation appears to have damaging effects during pregnancy as well as postnatally, maternal and child cash and/or nutritious foods transfers might be an effective means of preventing undernutrition throughout the first 1000 days (and potentially beyond), especially if they are combined with effective diet and nutrition-focused social behavior change communication[47,48] and if cash transfers are adjusted for inflation. Nutrition surveillance and multi-dimensional early warning systems warrant further investment in an era of greater food price volatility and more extreme weather events[49], as do programs designed to monitor and treat severe acute malnutrition[50].

Finally, since food inflation is clearly a major risk factor for various forms of child undernutrition, food policies, investments, and institutions should be aligned with recommendations made to reform national, regional and global food systems to achieve greater stability in food prices. Recommendations include scaling up investment in climate-smart agricultural R&D, new approaches to physical grain stocks or virtual stocks, closer regulation and analysis of biofuels policies, and stronger regulations on beggar-thy-neighbor trade policies such as export restrictions[51–53].

## Methods
### Data
To assess the impact of real food price shocks on child wasting and stunting we linked a large multi-country child-level DHS dataset[54] with national level FAO[40] data on separate consumer price indices for food and for all consumer goods and services. The DHS are highly standardized, nationally and subnationally representative, and contain a wide array of health, demographic, and socioeconomic indicators. The DHS Procedures and questionnaires for standard DHS surveys have been reviewed and approved by ICF Institutional Review Board (IRB), while country-specific DHS survey protocols are reviewed by the ICF IRB and typically by an IRB in the host country to ensure that the survey complies with the U.S. Department of Health and Human Services regulations for the protection of human subjects (45 CFR 46), while the host country IRB ensures that the survey complies with laws and norms of the nation.

Our DHS dataset comprises 130 surveys with anthropometric indicators for children 0–59 months of age in 44 LMICs surveyed between 2000 and 2021 (see Supplementary Table 3). Our criterion for country-year inclusion was any country with multiple DHS rounds that collected anthropometric indicators over the period of FAO CPI availability (2000 to the present). The dataset is representative of approximately 400 million under-5 children, around a quarter of them in India.

DHS data were used to calculate WHZ and HAZ scores relative to WHO reference standards for healthy breastfed children in multiple countries[55], wasting and stunting and severe wasting or stunting identifiers using the −2 and −3 standard deviations cut-offs, respectively. For wasting our key explanatory variable is the change in the ratio of the food CPI to the total CPI in the 3-month window preceding anthropometric measurement, while for stunting we measure the 9-month change during pregnancy (matching to the child's birth month), and 12-month changes when the child was 0–11 and 12–23 months[40]. The food and total CPIs are weighted indices of consumption baskets intended to represent typical consumption patterns of the population at large.

We also examined adequate child diet diversity, based on 24 h recall of 7 different food groups. Children who consumed four or more groups were classified as having an adequate diet. We use this indicator as a dependent variable to assess whether food inflation adversely affects child diet quality. We also used child diarrhea and fever in the previous 2 weeks to explore disease pathways.

The remaining variables in our analysis are control variables specified to minimize the bias of confounding factors. To employ a comparable measure of wealth across a wide range of countries we developed a simple classification of ownership of five assets (improved flooring, electricity, TV, fridge, and car/motorbike) and classified households as asset-poor if none of these were owned. We controlled for maternal education, three proxies designed to capture the continuum of maternal and child health care (antenatal care (% mothers who attended ≥4 visits in previous pregnancy), medical facility births and vaccinations (% children fully immunized for age), improved sanitation and water facilities, household demographics (teenage births, high fertility rates ≥4 children) as well as child sex and rural location. We used a DHS indicator of whether a household owned any farmland as an interaction variable, but only for those specific regressions since the indicator was only available for around 85% of the full sample. We also employed additional national-level controls for robustness checks, since these factors could influence food price changes but independently affect child health (see Supplementary Table 2)[56].

### Statistical analysis
Our analysis of these data was conducted in five steps.

First, we used the three-step weighting procedure for multi-country DHS analyses proposed in a recent paper on economic growth and child wasting[41], which renders regression coefficients representative of the DHS countries in any given sample. Supplementary Table 4 compares country sample sizes and under-5 populations to gauge the extent to which re-weighting is relevant.

Second, we use different descriptive analysis techniques to explore patterns and trends in the data. We used non-parametric local polynomial regressions with 95% confidence intervals to plot wasting and stunting by child age, and also to look at wasting-age patterns for gender, asset-poverty status, rural/urban location, and farm/nonfarm ownership within rural areas.

Third, we used weighted multivariate linear probability models to test the associations between real food price changes and wasting or stunting[41]. One feature of this model is that we interact food inflation with each country's national average wasting or stunting prevalence across surveys to allow the effect of inflation shocks to be linearly proportional to a country long-run undernutrition prevalence. This makes biological sense since populations with high undernutrition at baseline should be more vulnerable to negative shocks, but it also has a mathematical logic since a WHZ or HAZ distribution that is centered closer to a given wasting or stunting threshold should see larger absolute changes in undernutrition from a given shock. To ease

interpretation, we also re-scale the inflation indicators such that the coefficient can be interpreted as the proportional change in wasting risk from a 5% increase in the food/total CPI ratio. For a 3-month lag in real food inflation this amounts to approximately a two standard deviation increase.

Another feature of this model[41] is that it saturates the regressions with DHS controls, country fixed effects, and a series of interactions between region identifiers and three types of temporal factors: (1) long run time trends captured by 5-year time brackets; (2) month of survey variables to control for seasonality in wasting; and (3) child age in months variables to capture region-specific wasting-age dynamics. The regions were: the Sahel, the Horn of Africa, Western and Central Africa, and Eastern and Southern Africa, South Asia, South-East Asia, Middle East and North Africa, Eastern Europe and Central Asia and Latin America and the Caribbean.

With the control variables outlined above, the linear probability model for wasting takes the form:

$$N_{i,c,r,t} = \beta_0 + \beta_g \bar{w}_{c,r} f_{c,r,t-3} + \boldsymbol{\beta_X X_{i,c,r,t}} + \boldsymbol{\beta_C C} + \boldsymbol{\beta_A A.R} + \boldsymbol{\beta_S S.R} + \boldsymbol{\beta_T T.R} + \varepsilon_{i,c,r,t}$$
(1)

This equation states that wasting ($N$) for child $i$ in country $c$ and region $r$ at time $t$ is modeled as a function of real food rice changes over three months prior to the month of wasting measurement ($f_{t-3}$) interacted with average wasting prevalence across all rounds ($\bar{w}$). We note that mean wasting ($\bar{w}$) refers to means of each specific wasting indicator (wasting or severe wasting), depending on which is specified on the left-hand side of Eq. (1). The remaining variables in Eq. (1) include a vector of control variables from the DHS ($X$), country fixed effects ($C$), and the three types of region-specific temporal effects (child age effects ($A.R$), seasonality effects ($S.R$) and trend effects ($T.R$)). Since the specification includes country fixed effects, the coefficient on food inflation represents the elasticity of wasting risks with respect to deviations of real food price changes from their long-term average change, which one could therefore think of as food inflation shocks. Standard errors ($\varepsilon$) are clustered at the country level for the calculation of 95% confidence intervals (CIs). All analysis was conducted in STATA™ Version 17.

Although all our regressions follow the basic structure of Eq. (1), the wasting regressions stratified results by age (0–11 months, 12–23 months, and 24–59 months), and then look at the 0–5 month window to test for potential birthweight effects. We then introduced interaction terms with the asset-poor indicator, child gender, urban location, and farm ownership (in the rural sample only). We also tested sensitivity to other national level controls that could be associated with food inflation but independently predict wasting.

In a fourth step we adapt Eq. (1) to model stunting status for children 24–59 months as a function of inflation shocks from: (1) conception to birth (9 months); (2) birth to age one year (11–23 months); and (3) age one to aged two years (12–23 months). Food inflation variables are again interacted with a country's mean prevalence of stunting and rescaled to represent 5 percent changes in the food CPI/total CPI ratio.

The final step in our analysis we modeled the risk of inadequately diverse diets, measured as a failure to consume at least four of the seven possible food groups in the past 24 h, as a function of real food price changes over the 12 months prior to dietary measurement. We also tested the more unlikely hypothesis that real food inflation might influence nutrition through a disease pathway, using child diarrhea or fever in the past 2 weeks as the dependent variable. In all three regressions, the control variables remain the same as Eq. (1).

## Reporting summary

Further information on research design is available in the Nature Portfolio Reporting Summary linked to this article.

## Data availability

The Demographic Health Survey data used in this analysis are available under restricted access as per the data access policies of the DHS Program, though access request after registering on the DHS website: https://dhsprogram.com/data/Using-Datasets-for-Analysis.cfm. The FAO consumer food price data are publicly available at https://www.fao.org/faostat/en/#data/CP. The analysis in this paper also conforms to the Guidelines for Accurate and Transparent Health Estimates Reporting: the GATHER statement[57]. Source data are provided with this paper.

## Code availability

Stata™ v17 code for replicating this study is publicly available[58].

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

## Acknowledgements

D.H. was funded by the Micronutrient Forum, USA, on behalf of the Standing Together for Nutrition (STfN) consortium through a grant from Global Affairs Canada; and the Food Prices for Nutrition project funded by The Bill and Melinda Gates Foundation (BMGF) and the United Kingdom government's Foreign, Commonwealth & Development Office. We also thank BMGF for supporting an earlier project (*Advancing Research on Nutrition and Agriculture*) that helped build and maintain the multi-country Demographic Health Surveys (DHS) dataset used for this analysis. This work was also part of the CGIAR Research Initiative on Sustainable Healthy Diets through Food Systems Transformation (SHiFT) led

by the International Food Policy Research Institute (IFPRI) and the Alliance of Bioversity International and the International Center for Tropical Agriculture (the Alliance), in partnership with Wageningen University and Research (WUR) and with contributions from the International Potato Center (CIP). We would like to thank all funders who supported this research through their contributions to the CGIAR Trust Fund: https://www.cgiar.org/funders/. The authors also extend their gratitude to Kalle Hirvonen, Channing Arndt, Harold Alderman, and the STfN steering committee for constructive feedback. We also thank Wahid Quabili for valuable assistance in preparing the data.

## Author contributions

D.H. and M.R. conceptualized the study. D.H. conducted the data collection and empirical analysis. Both D.H. and M.R. contributed to the writing.

## Competing interests

The authors declare no competing interests.
