## [Peer Review File · Nature Communications]

nature portfolio

Peer Review FileReviewer comments, first round

Reviewer #1 (Remarks to the Author):

Thanks for the invitation to review the manuscript, "Food inflation and child undernutrition in low and middle income countries." This paper examines the relationship between recent and/or early-life exposures to food inflation and childhood wasting and stunting, tests for variation in these effects across relevant sub-populations, and provides suggestive analyses of potential dietary and health pathways. In my view, the paper addresses an important topic and is generally well-written and well-executed. I would expect the paper to be of interest to the wide audience of Nature Communications. With this in mind, I have some questions and comments, listed below in the order they were encountered (generally).

- The paper relates strongly to the authors' recent paper in this journal on economic shocks and child wasting. While this is not necessarily a problem, it would be helpful to see some discussion in either the main text or supplemental materials about how these papers differ beyond their examination of different forms of macro-economic stress.
- Line 34: should this be the "21st century"?
- Line 142-3: the first two parts of this sentence seem a bit contradictory; is one part referring to newborns and the other part to slightly older children?
- Figure 2: instead of mentioning the Stata command, it might be more informative to provide some details about the regression (e.g., how is the approach to smoothing defined?)
- I was not clear whether the key exposure term is food inflation or real increases in food costs. Perhaps this reflects my background in demography instead of economics, but my understanding is that inflation is typically defined as an increase in nominal costs while real increases in a given cost reflect changes net of inflation. It would be helpful to clarify this issue. Relatedly, I found the exposure term (food CPI : total CPI) to have a non-intuitive interpretation, so would recommend a "plain language" definition of this variable and how it should be interpreted. In my view this will increase the likelihood that readers interpret the results correctly.
- Did the authors test for any non-linearities in inflation effects? I am particularly interested in whether we can reasonably assume that deflationary pressures would have symmetrical effects as inflation.
- Are linear probability models acceptable for modeling all of these outcomes given how rare some of them (e.g., wasting, severe wasting) are? For example, do the results of the models generate predicted probabilities <0 within the range of the data?
- The notes for the figures and tables should be checked. For example, the note of Figure 4 refers to parentheses that are not in the figure, to a sample, of just 1.271 children, and to the representativeness of the sample vis-à-vis a sample instead of a target population. There are similar issues in other captions.
- Line 261: I agree that the results suggest additive effects, but this reflects the structure of the model. For example, interactive effects also could have been observed if the models included the necessary multi-way interaction terms.
- Line 290: the authors state that they do not observe the *same* interactions for stunting as they do for wasting, but it would be helpful to note if *any* notable interactions are observed.
- Line 310+: given that children as young as 6 months are included in the sample, is there any concern the results could be confounded by systematic differences in the prevalence of breastfeeding (which could affect MDD)?
- Lines 322 and 325: reported age ranges of 6-23 months and 6-24 months are contradictory.
- I would recommend a supplemental analysis of diarrhea and fever using the same sample used for MDD so the results can be compared.
- Line 375: the analyses of mortality are possible with these DHS data. Perhaps this is beyond the scope of this single paper, but these selection effects could be evaluated empirically.
- The two paragraphs between lines 370 and 394 include some repetition of the same points from paragraph to paragraph.
- Given that India accounts for such a large share of the sample, it would be helpful to assess whether the results are driven by that one country, or any other countries that might be influential.

- Are the variables reported around lines 442+ individual or country-level variables? They appeared to be individual level but the reference to percentages was not clear.
- Line ~465: are the results sensitive to interacting with the country's mean WHZ or HAZ instead of the wasting or stunting rates?
- Line 494: I would recommend just stating the variables in this text instead of listing in a later appendix table.
- Are the same temporal trends controlled for in the stunting models as the wasting models? I would be concerned that it may not be necessary to account for seasonality when modeling stunting; and either way it would be helpful to clarify (e.g., around lines 496-500).
- Table S2: the GDP growth rates seem implausibly small (assuming I am reading this correctly).

Reviewer #2 (Remarks to the Author):

This manuscript provides a timely and innovative use of existing data to interrogate the relationship between food price inflation and childhood undernutrition. The DHS data are generally of high quality and available for many low and lower-middle income countries, though the lack of survey data from upper-middle income countries, is a limitation for the claimed implications for "middle income countries". The use of the relative increase in food CPI vs total CPI is appropriate for these analyses. However, there may be limitations that should be noted regarding variations and difficulties in measurements among countries. Also the CPI is more oriented to urban consumption than rural and this may have implications for their finding of more effect in rural populations.

The lengthy descriptive section on wasting and stunting prevalence uses previously published methods of analysis and finds patterns that have been widely published. They have grouped countries into subregions of Africa and Asia, but have not provided justification for these groupings, or uncertainty information for each. The main analyses related to food price inflation are not possible for these subregions because of limited DHS data so the descriptive figures are not directly relevant. It may be better to put these analyses in an appendix to describe the data being used but not present as novel results.

For the main regression results the time periods before assessment of stunting and wasting in the surveys is appropriate but DHS surveys may be conducted over a 6 month period or longer in a country. Was it possible to use the survey date for each child to calculate this exposure period? If not what are the implications for the analyses, especially for wasting? The results for wasting and stunting are plausible, albeit with wide confidence intervals.

Their attempts to assess diet and infections as mechanisms to explain the relationship of food price inflation and wasting and stunting are problematic. The minimum dietary diversity metric is a poor indicator of diet quality including caloric adequacy. A tenuous correlation to intake of micronutrients has been claimed in previous literature but not well demonstrated. The authors cite an inaccessible 16 year old report for this claim. They also claim a relationship with energy intake, which is very questionable, citing their own book chapter. The survey reports of diarrhea and fever in the past two weeks are not meaningful measures on infectious illness prevalence. They do not provide valid evidence for the effect of infection on growth. The effect of food prices on diet diversity can contribute to the full picture, but more caution is needed regarding this being a mechanism for the effects on growth. If they want to investigate MDD as a mechanism more fully they should do a mediation analysis for the effects on wasting and stunting.

A minor point is that in the title and throughout it seems that they should use food price inflation rather than food inflation.

Reviewer #3 (Remarks to the Author):

This paper investigates the relationship between food price shocks and child nutritional status

across 44 developing countries using comparable Demographic and Health Surveys (DHS). I found the empirical exercise useful, subject to some caveats.

Comments:

1) To an economist, the term "inflation" applied to food prices is an oxymoron. Inflation refers to a general increase in the price level whereas here the authors are talking about spikes in the relative price of food. I would strongly recommend using the term "food price shocks" consistently throughout the paper instead of inflation. And "shock" is preferable to "spike" because, as shown in Fig. 3, the relative price of food in the data both rises and falls so that Δp looks close to mean zero. By the same token, testing for impacts of "total inflation" seems superfluous, as a pure inflation should raise nominal income by the same percentage as expenditures implying no change in real income.

2) Since, as just noted, the data encompass both food price spikes and drops in roughly equal proportion, why not test whether these have symmetrically opposite effects on child nutritional status as would be implied if the mechanism at play is movement along the food demand curve? If symmetry cannot be rejected, then food price volatility in and of itself may not be too harmful (would depend on possibility of catch-up growth), which might nuance the policy implications regarding food price stabilization measures.

3) In the regression model given by equation (1), the change in food prices is interacted with the average wasting prevalence across rounds for the country; there is no separate linear term in the food price change. The purpose of this peculiar specification is to correct for an inconvenient feature of the linear probability model, which is that the marginal effect of a shock is constant across different wasting probabilities. Meanwhile, however, all the other controls, including time trends, enter only linearly. Why is only one covariate, the food price change, privileged with this "correction"? This inconsistency in specification could potentially weaken the explanatory power of the controls and correspondingly strengthen the effect of price shocks. There are two ways around this problem: (1) interact all covariates with mean wasting, which is essentially the same thing as using child wasting normalized by mean wasting for the country as the dependent variable with a heteroskedasticity correction (i.e., allowing the residual variance to depend on mean wasting); or, simpler yet, (2) move to a probit or logit specification for the wasting/stunting probability.

4) Some comment is perhaps warranted about the validity/accuracy of anthropometric measurements for young infants (0-12) months in developing country settings. Weight-for-height is also notoriously noisy since its calculation involves the ratio of two error-ridden measures. I wonder how much of the age pattern in the Sahel, for instance, is due to noise (Fig. 2A); adding confidence intervals would help. Why not use weight-for-age as a measure of acute malnutrition?

REVIEWER COMMENTS

Reviewer #1 (Remarks to the Author):

Thanks for the invitation to review the manuscript, “Food inflation and child undernutrition in low and middle income countries.” This paper examines the relationship between recent and/or early-life exposures to food inflation and childhood wasting and stunting, tests for variation in these effects across relevant sub-populations, and provides suggestive analyses of potential dietary and health pathways. In my view, the paper addresses an important topic and is generally well-written and well-executed. I would expect the paper to be of interest to the wide audience of Nature Communications. With this in mind, I have some questions and comments, listed below in the order they were encountered (generally).

Sincere thanks to the reviewer for very detailed and thoughtful comments – the paper is genuinely improved after addressing these comments, so the care taken is much appreciated.

- The paper relates strongly to the authors’ recent paper in this journal on economic shocks and child wasting. While this is not necessarily a problem, it would be helpful to see some discussion in either the main text or supplemental materials about how these papers differ beyond their examination of different forms of macro-economic stress.

Thank you for this comment. In the introductory paragraph we cite our earlier paper noting “A number of studies link climate shocks to wasting or stunting among pre-schoolers,²⁶⁻²⁸ or even short stature in adulthood,^{29,30} while other studies examine the impacts of macroeconomic crises on these nutrition outcomes.^{31,32} Such crises might often involve food inflation, but are also characterized by rising unemployment and declines in nominal incomes, making it difficult to isolate the specific impacts of food inflation on nutrition outcomes.” And then in the results section we now note the following:

“We also note that the correlation between 3-month real food price changes and economic growth is 0.07, though it is not statistically significant; hence this measure does not seem closely linked to the kind of economic growth shocks analyzed in another recent paper on the macroeconomic determinants of child malnutrition.³²”

- Line 34: should this be the “21st century”?

Yes! Thank you for pointing this out.

- Line 142-3: the first two parts of this sentence seem a bit contradictory; is one part referring to newborns and the other part to slightly older children?

We have re-phrased this sentence.

- Figure 2: instead of mentioning the Stata command, it might be more informative to provide some details about the regression (e.g., how is the approach to smoothing defined?)

Thank you for this suggestion. We have now revised the notes to the Figure as follows:

“Source: Authors’ estimates from DHS data using the `lpoly` command in STATA™. `lpoly` performs a kernel-weighted local polynomial regression of wasting or stunting status on child age (i.e. including polynomials of child age. See Section 4 on Methods and Materials for more details, as well as <https://www.stata.com/manuals13/rlpoly.pdf>.”

- I was not clear whether the key exposure term is food inflation or real increases in food costs. Perhaps this reflects my background in demography instead of economics, but my understanding is that inflation is typically defined as an increase in nominal costs while real increases in a given cost reflect changes net of inflation. It would be helpful to clarify this issue. Relatedly, I found the exposure term (food CPI : total CPI) to have a non-intuitive interpretation, so would recommend a “plain language” definition of this variable and how it should be interpreted. In my view this will increase the likelihood that readers interpret the results correctly.

We think the reviewer raises a good point, and in the first draft we struggled to come up with the best terminology for a multi-disciplinary audience, but we agree that “real inflation” is close to a contradiction in terms. We now refer just to real food price changes throughout, and some text to describe the intuition around the specific foodCPI/totalCPI ratio: if positive it just means food prices are increasing faster than the full set of consumer prices. Please note that we have also changed the title of the paper accordingly to: “Rising food prices and child undernutrition in low and middle income countries”. When we introduce the measure we also write:

“We term changes in this food CPI/total CPI ratio as “real food price increases”, and the indicator measures whether food inflation has outpaced the inflation rate of all consumer goods and services. We use this real price change indicator, rather than nominal food price changes (food inflation), because we hypothesize that it is relative food price changes that are an economic risk for the poor, who spend large share of their budgets on food, and have few means of adapting to higher prices in the short run.”

- Did the authors test for any non-linearities in inflation effects? I am particularly interested in whether we can reasonably assume that deflationary pressures would have symmetrical effects as inflation.

This is a very good question and we have now added supplementary Figure S7 and S8 and relevant text in the main paper. There are not strong indications of any asymmetries or non-linearities, and indeed it seems that declines in real food prices predict reductions in wasting risks. We have added a new subsection devoted to this symmetry issue and we write:

“Figure S7 reports results of testing for asymmetric associations between positive and negative real food price changes and child wasting. Relative to negative or no food price change, increases in the real price of food elevate the risk of wasting, as expected (Panel A, Figure S7), but we also find that decreases in the price of food significantly reduce the risk of wasting. Likewise for stunting (Figure S8) we find that real food price increases in the prenatal period are associated with a significantly elevated risk of stunting among children 24-59 months, while the association for food price decreases is negative as expected, but marginally insignificant at the 10% level ($p=0.12$). Hence there are few signs of non-linear or asymmetric associations between food price changes and stunting or wasting risks.”

Figure S7. Testing for asymmetries in the estimated effects of real food price changes on wasting: weighted multivariate linear probability coefficients of moderate/severe wasting as a function of either increases in real food price changes (Panel A) or decreases in real food prices (Panel B)

Panel A: Increases in the real price of food

Panel B: Decreases in the real price of food

Notes: 95% confidence intervals based on standard errors clustered at the country level are reported in parentheses. Panel A reports results from a regression that specifies the impact of increases in the real price of food relative to a dummy variable equal to zero for all negative food price changes or no price change. Panel B conversely reports results from a regression that specifies the impact of decreases in the real price of food (in absolute terms) relative to a dummy variable equal to zero for all positive food price changes or no price change. Regressions are weighted (See Methods and Materials). However, the results are not sensitive to inclusion/exclusion of either food price change measure. The model also incorporates an extensive set of controls described in the Methods and Materials, including DHS variables, and various temporal effects and country fixed effects.

•Are linear probability models acceptable for modeling all of these outcomes given how rare some of them (e.g., wasting, severe wasting) are? For example, do the results of the models generate predicted probabilities <0 within the range of the data?

We also faced this question with regard to our previous paper, so have considered the issue in some depth. Econometricians tend to have a different view from public health/nutrition statisticians, with a preference for linear probability methods, for various reasons outlined here:

<https://blogs.worldbank.org/impacetevaluations/whether-to-probit-or-to-probe-it-in-defense-of-the-linear-probability-model>

Particularly telling is a quote from arguably the leading econometric textbook from Woolridge:

“...If the main purpose is to estimate the partial effect of [the independent variable] on the response probability, averaged across the distribution of [the independent variable], then the fact that some predicted values are outside the unit interval may not be very important.”

Moreover, with fixed effects in the specification, PROBIT models are not recommended, and often will not converge to provide estimates, so it is not really an option in this instance.

•The notes for the figures and tables should be checked. For example, the note of Figure 4 refers

to parentheses that are not in the figure, to a sample, of just 1.271 children, and to the representativeness of the sample vis-à-vis a sample instead of a target population. There are similar issues in other captions.

Thank you for pointing this out. We have now gone through all the main text and supplementary tables and figures and made corrections.

•Line 261: I agree that the results suggest additive effects, but this reflects the structure of the model. For example, interactive effects also could have been observed if the models included the necessary multi-way interaction terms.

A good point – we have noted in the text that the model specification implies additive effects, but with so many variables we are reluctant to add additional multi-way effects, especially as the model here is only supposed to be illustrative of these heterogeneities. We now add “(although it is also possible to specify further interaction terms).”

•Line 290: the authors state that they do not observe the *same* interactions for stunting as they do for wasting, but it would be helpful to note if *any* notable interactions are observed.

We have clarified more precisely how the results differ. Thank you for pointing out this ambiguity. “However, there are fewer significant interaction effects in the stunting model, and the only interaction that is highly significant suggests that food inflation in the prenatal period has a large risk for stunting among urban children compared to rural children (Supplement Table S7), whereas the opposite was the case with wasting (Table 1).”

•Line 310+: given that children as young as 6 months are included in the sample, is there any concern the results could be confounded by systematic differences in the prevalence of breastfeeding (which could affect MDD)?

We ran regressions on a 12-23 month sample, the 6-23 month sample controlling for breastfeeding, the 6-23 month sample for breastfed children only and 6-23 month children not breastfed. The point estimates remain unchanged throughout, except the 6-23 month children not breastfed, where the coefficient drops slightly and is only significant at the 10% level. However, only 20% of our 6-23 month sample is not breastfed, so the last result is likely just imprecise because of a smaller sample, and also a richer sample (non-breastfed children are from households that are substantially better off in terms of assets, and more likely to be urban).

•Lines 322 and 325: reported age ranges of 6-23 months and 6-24 months are contradictory.

Thanks for spotting this – now corrected

•I would recommend a supplemental analysis of diarrhea and fever using the same sample used for MDD so the results can be compared.

We have just made the main figure in the text for 6-23 month children only. In the table notes we report that the diarrhea and fever results are the same if the 0-59m sample is used.

•Line 375: the analyses of mortality are possible with these DHS data. Perhaps this is beyond the scope of this single paper, but these selection effects could be evaluated empirically.

We make note of this in the limitations, but addressing selection effects would indeed be very challenging and beyond the scope of the present paper and the dataset we use.

•The two paragraphs between lines 370 and 394 include some repetition of the same points from paragraph to paragraph.

Thank you – we have edited and streamlined this. There was indeed some duplication.

•Line ~465: are the results sensitive to interacting with the country's mean WHZ or HAZ instead of the wasting or stunting rates?

This is not really a valid exercise, because the coefficient would not be interpreted as an elasticity (proportional risk) if we interacted with WHZ, so in this particular instance we have not followed the reviewer's suggestion.

•Line 494: I would recommend just stating the variables in this text instead of listing in a later appendix table.

We have now added some text describing these variables.

•Are the same temporal trends controlled for in the stunting models as the wasting models? I would be concerned that it may not be necessary to account for seasonality when modeling stunting; and either way it would be helpful to clarify (e.g., around lines 496-500).

Yes, they are the same, and indeed it may not be necessary to account for seasonality in stunting, but it will also do no harm, and seasonality in stunting is at least conceptually possible. We have clarified that the controls are the same.

•Table S2: the GDP growth rates seem implausibly small (assuming I am reading this correctly).

Thank you for spotting this. There was an error for the GDP variable and also two of the other variables were in proportions instead of percentages.

Reviewer #2 (Remarks to the Author):

This manuscript provides a timely and innovative use of existing data to interrogate the relationship between food price inflation and childhood undernutrition. The DHS data are generally of high quality and available for many low and lower-middle income countries, though the lack of survey data from upper-middle income countries, is a limitation for the claimed implications for "middle income countries".

This is a good point. We have now made more note on the limitations of representativeness. In terms of countries/regions that still have relatively high wasting and stunting, the coverage is not so good for South-East Asia (e.g. Indonesia). We now write "The DHS is not representative of all LMICs, with a lack of data for South-East Asia – where wasting and stunting are still relatively high – perhaps being the most obvious gap."

The use of the relative increase in food CPI vs total CPI is appropriate for these analyses. However, there may be limitations that should be noted regarding variations and difficulties in measurements among countries. Also the CPI is more oriented to urban consumption than rural and this may have implications for their finding of more effect in rural populations.

An excellent point – we hadn't thought of this. We have added text in the results section to point out that this problem could potentially explain the somewhat surprising finding on rural-urban differences in "impacts", and also made a note in the limitations section.

". . . It also possible that the national price indices we use less accurately reflect "on the ground" prices in remote rural areas."

"There are also concerns that consumer price surveys are urban biased,⁴⁶ so the CPI measures we used may not capture local food price shocks in rural areas very accurately and could affect our tests for heterogeneous tests of impacts across rural and urban areas."

The lengthy descriptive section on wasting and stunting prevalence uses previously published methods of analysis and finds patterns that have been widely published. They have grouped countries into subregions of Africa and Asia, but have not provided justification for these groupings, or uncertainty information for each. The main analyses related to food price inflation are not possible for these subregions because of limited DHS data so the descriptive figures are not directly relevant. It may be better to put these analyses in an appendix to describe the data being used but not present as novel results.

We considered this suggestion and indeed removed the Figure in question to the supplement, but it seemed to disrupt the flow of the paper, because the patterns of stunting and wasting by region and by child age are quite important for the readers' understanding of our regression methods; for example, why we look at stunting for kids 24-59 months and not 0-59 months, and why we also break up wasting by age ranges, and why we do these interactions with mean wasting and stunting rates. However, we agree that the results are not highly novel and the text was too long, so we have considerably shortened the text in question to just focus on a few key takeaways from the graph in question. This makes the results section more concisely written up.

For the main regression results the time periods before assessment of stunting and wasting in the surveys is appropriate but DHS surveys may be conducted over a 6 month period or longer in a country. Was it possible to use the survey date for each child to calculate this exposure period? If not what are the implications for the analyses, especially for wasting? The results for wasting and stunting are plausible, albeit with wide confidence intervals.

Yes, indeed, we actually exploit the fact that the DHS was conducted in different months to get more variation in exposure to food price changes and we match food prices to the DHS indicator of the month in which a child was measured for anthropometrics. This was the method followed in Arndt et al.'s study of Mozambique where they used a survey that was conducted during both a high food price period and a low/falling food price period. So we use the same logic. That also explains why we use a 3 month lag – which gets us more variation in food prices than longer lags. When we use longer lags we get roughly the same point estimates but less precision.

The point on wide confidence intervals is well noted, and not surprising given that we rely on national food price data (potentially urban biased as the reviewer noted) and that there may be unspecified heterogeneity in impacts. For stunting and wasting there is also potentially issues of measurement error, including child age recall problems. We have now made reference to these measurement issues in the limitations and added reference to some of the problems with anthropometric measurement, which was also raised by Reviewer 3.

Their attempts to assess diet and infections as mechanisms to explain the relationship of food price inflation and wasting and stunting are problematic. The minimum dietary diversity metric is a poor indicator of diet quality including caloric adequacy. A tenuous correlation to intake of micronutrients has been claimed in previous literature but not well demonstrated. The authors cite an inaccessible 16 year old report for this claim. They also claim a relationship with energy intake, which is very questionable, citing their own book chapter. The survey reports of diarrhea and fever in the past two weeks are not meaningful measures on infectious illness prevalence. They do not provide valid evidence for the effect of infection on growth. The effect of food prices on diet diversity can contribute to the full picture, but more caution is needed regarding this being a mechanism for the effects on growth. If they want to investigate MDD as a mechanism more fully they should do a mediation analysis for the effects on wasting and stunting.

We agree with the reviewer that perhaps we should be more cautious on this front, but we also think the dietary diversity metrics clearly have value, especially in the absence of any other dietary data in the DHS. Moreover, it is hard to see why the reviewer thinks dietary diversity is a poor indicator of diet quality – it is perhaps fair to say it is “rough” or only a proxy (the term we use in the text), but there are a huge number of studies showing that simple child diet diversity indicators are positively associated with micronutrient intake. We have now added more up to date systematic review on this evidence:

Molani-Gol, R., Kheirouri, S., & Alizadeh, M. (2023). Does the high dietary diversity score predict dietary micronutrients adequacy in children under 5 years old? A systematic review. *Journal of Health, Population and Nutrition*, 42, 2.

We dropped the reference to calorie intake, as the evidence on that front is perhaps less extensive.

There is also plenty of literature showing strong associations between dietary diversity indicators for children and child HAZ or stunting, and that in spite of the very short recall.

The reviewer claims the 16 year old report that first validated the metric is inaccessible – this is not true. A quick google search showed it here, and on other websites:

<https://www.fantaproject.org/research/indicators-dietary-quality-intake-children>

The paper already noted the problems with diarrhea and fever measurement, but we have further edited the text to make that limitation very clear.

Finally, a mediation analysis is not really appropriate in this setting. Although there is a correlation between stunting and child diet diversity, for example, we cannot say that this is a causal association or an unbiased estimate. The short recall period on dietary diversity means that the association is almost certainly attenuated by the fact that long-term dietary diversity (quality) influences stunting. This is noted in a paper below:

Thorne-Lyman, A., Spiegelman, D., & Fawzi, W. W. (2014). Is the strength of association between indicators of dietary quality and the nutritional status of children being underestimated? *Maternal & Child Nutrition*, 10, 159-160.

We would only favour a mediation analysis when measurement error in the mediation variable is not really an issue (but the reviewer emphasizes the limitations of diet diversity data). We have now added that to the limitations section.

“The DHS indicator of child diet diversity has been shown to be correlated with micronutrient intake of children,⁴⁴ but has a short recall period that could be problematic for this and other kinds of analyses.⁴⁸ The DHS does not provide information on maternal diets or weight gain during pregnancy. The DHS indicators of recent symptoms of illness (diarrhea, fever) are also less than ideal for exploring potential health impacts from food price changes, and we should be cautious interpreting the lack of an association as strong evidence of lack of impact of food price change on child health.”

In summary, we agree that we should be more cautious on this evidence on potential mechanisms, but we don't agree that this analysis should be discarded because dietary diversity indicators are too poor a proxy of dietary quality. The literature doesn't seem to support that view.

A minor point is that in the title and throughout it seems that they should use food price inflation rather than food inflation.

Thank you for this point, which the other reviewers also noted. We now refer to food price changes or increases throughout, including in the new title.

Reviewer #3 (Remarks to the Author):

This paper investigates the relationship between food price shocks and child nutritional status across 44 developing countries using comparable Demographic and Health Surveys (DHS). I found the empirical exercise useful, subject to some caveats.

Comments:

1) To an economist, the term “inflation” applied to food prices is an oxymoron. Inflation refers to a general increase in the price level whereas here the authors are talking about spikes in the relative price of food. I would strongly recommend using the term “food price shocks” consistently throughout the paper instead of inflation. And “shock” is preferable to “spike” because, as shown in Fig. 3, the relative price of food in the data both rises and falls so that Δp looks close to mean zero.

We agree with this point and it was raised by other reviewers also. We also considered the term shock, but perhaps thought it more neutral and accurate just to say food price changes. For one thing, some readers may interpret shock as a food price increase, but we are looking here at increases and decreases, some small, some large. In the text, however, we do point out that the changes could be interpreted as shocks.

By the same token, testing for impacts of “total inflation” seems superfluous, as a pure inflation should raise nominal income by the same percentage as expenditures implying no change in real income.

While we agree that the null association between total inflation and child nutrition outcomes is not that surprising from an economic perspective, it's not obvious that pure inflation should immediately raise nominal income by the same percentage as prices (the reviewer writes expenditures, but we are not sure why – perhaps “prices” was intended). Right now, in several high-inflation countries we are also observing wages to be slow to adjust to rising prices. We have now added two references on this:

Easterly, W. H., & Fischer, S. (2001). Inflation and the poor. *Journal of Money, Credit and Banking*, 33, 160-178.

Headey, D. D., Bachewe, F. N., Marshall, Q., Raghunathan, K., & Mahrt, K. 2023. Food prices and the wages of the poor: A low-cost, high-value approach to high-frequency food security monitoring. IFPRI Discussion Paper 2174. Washington DC.

2) Since, as just noted, the data encompass both food price spikes and drops in roughly equal proportion, why not test whether these have symmetrically opposite effects on child nutritional status as would be implied if the mechanism at play is movement along the food demand curve? If symmetry cannot be rejected, then food price volatility in and of itself may not be too harmful (would depend on possibility of catch-up growth), which might nuance the policy implications regarding food price stabilization measures.

Thank you for this suggestion, which was also note by Reviewer 1. We have now added two figures in the supplement testing for asymmetric effects, but it appears we can reject asymmetry as food price declines also seem beneficial for nutrition, on average. We also think your

hypothesis that volatility is not bad is an interesting one. That is perhaps supported by the stunting results to some extent, because those “impacts” are fairly modest, and that may be because periods of rising food prices are typically followed by periods of declining food prices. For wasting, something similar might be taking place (weight loss and then weight gain as prices fall) but of course the fear there is that wasting is a major risk factor for morbidity and mortality. We have reflected your hypothesis in the discussion section now – good thought, for sure. The text we added is as follows and the asymmetry tests are below that:

“We also note that while our study implies that increases in food prices elevate the risk of wasting and stunting, the modelling results also imply that declines in food prices reduce the risk of wasting and stunting (we find no evidence of asymmetric effects between increases and decreases in real food prices). From that, one might conclude that pure food price volatility is harmless in net terms, but we caution against that conclusion. The benefits of falling food prices may explain why we observe relatively modest impacts of food prices increases on stunting (i.e. price increases may be followed by price declines that facilitate some catch-up growth), but even short run increases in wasting can substantially increase the risk of mortality in young children, and it is well known that a variety of nutritional insults *in utero* and early childhood can have longer term impacts on growth, cognitive development, educational attainment and adult health, as we noted in our introduction.^{26-29”}

Figure S7. Testing for asymmetrical associations between real food price changes and wasting: weighted multivariate linear probability coefficients of moderate/severe wasting as a function of either increases in real food price changes (Panel A) or decreases in real food prices (Panel B)

Panel A: Increases in the real price of food

Panel B: Decreases in the real price of food

Notes: 95% confidence intervals based on standard errors clustered at the country level are reported in parentheses. Panel A reports results from a regression that specifies the impact of increases in the real price of food relative to a dummy variable equal to zero for all negative food price changes or no price change. Panel B conversely reports results from a regression that specifies the impact of decreases in the real price of food (in absolute terms) relative to a dummy variable equal to zero for all positive food price changes or no price change. Regressions are weighted (See Methods and Materials). However, the results are not sensitive to inclusion/exclusion of either food price change measure. The model also incorporates an extensive set of controls described in the Methods and Materials, including DHS variables, and various temporal effects and country fixed effects. The sample size is 1.27 million children in 44 LMICs.

Figure S8. Testing for asymmetrical associations between real food price changes and stunting: weighted multivariate linear probability coefficients of moderate/severe stunting as a function of either increases in real food price changes or decreases in real food prices

Notes: 95% confidence intervals based on standard errors clustered at the country level are reported in parentheses. The first coefficient reports results from a regression that specifies the impact of increases in the real price of food relative in the prenatal period to a dummy variable equal to zero for all negative food price changes or no price change. The second coefficient conversely reports results from a regression that specifies the impact of decreases in the real price of food (in absolute terms) relative to a dummy variable equal to zero for all positive food price changes or no price change. Regressions are weighted (See Methods and Materials). However, the results are not sensitive to inclusion/exclusion of either food price change measure. The model also incorporates an extensive set of controls described in the Methods and Materials, including DHS variables, and various temporal effects and country fixed effects. The full sample includes 694,673 children in 44 LMICs.

3) In the regression model given by equation (1), the change in food prices is interacted with the average wasting prevalence across rounds for the country; there is no separate linear term in the food price change. The purpose of this peculiar specification is to correct for an inconvenient feature of the linear probability model, which is that the marginal effect of a shock is constant across different wasting probabilities. Meanwhile, however, all the other controls, including time trends, enter only linearly. Why is only one covariate, the food price change, privileged with this “correction”? This inconsistency in specification could potentially weaken the explanatory power of the controls and correspondingly strengthen the effect of price shocks. There are two ways around this problem: (1) interact all covariates with mean wasting, which is essentially the same thing as using child wasting normalized by mean wasting for the country as the dependent variable with a heteroskedasticity correction (i.e., allowing the residual variance to depend on mean wasting); or, simpler yet, (2) move to a probit or logit specification for the wasting/stunting probability.

Good question. Our main rationale for introducing the interaction with the mean of the dependent variable is that food price shock impacts in a low-wasting country are not likely to be the same as they are in a high-wasting country, so the interaction allows proportionality in

impacts and an elasticity interpretation. We only applied it to the coefficient on food prices (or in the previous study, the coefficient on economic growth) because this indicator is measured at the national level whereas all the DHS controls are measured at the child, maternal or household level. It is possible that the impacts of low education and household wealth vary by low and high wasting countries, but that heterogeneity is not likely to influence the results for food inflation because family-level DHS variables are just never going to be strongly correlated with a national food price change indicator, and they are really specified as a matter of form as standard controls, and because a few of the variables are used in interaction terms. The fact that DHS unit level variables are not strong controls for a national level indicator like food price changes is why we test robustness to other national controls (economic growth, conflict, broad money growth, food production growth, etc.) which are interacted with average wasting or stunting levels for precisely the reasons that the reviewer is concerned about (that the interaction between wasting and food prices may impart some kind of advantage for the coefficient estimation).

To assure the reviewer that our results are not influenced by the decision to only interact food prices and country-level wasting we show some additional regressions below where we first drop all the DHS control variables from the model and then interact all the controls with the mean of a country's wasting prevalence. The results are very robust, but this is what we'd expect precisely because the family level control variables and the food price shocks are not strongly correlated with each other, especially once we control for country fixed effects.

Here are the result in the main text with the DHS controls that are not interacted with mean wasting prevalence

And now results without DHS controls. As you see, the DHS controls don't have much influence at all.

And now results where all the DHS controls are interacted with a country's average wasting prevalence. The point estimates are virtually identical. So the reviewer has raised a valid concern, but in practice the choice makes no difference to the estimates of interest.

4) Some comment is perhaps warranted about the validity/accuracy of anthropometric measurements for young infants (0-12) months in developing country settings. Weight-for-height is also notoriously noisy since its calculation involves the ratio of two error-ridden measures. I wonder how much of the age pattern in the Sahel, for instance, is due to noise (Fig. 2A); adding confidence intervals would help.

This is a valid point. WHZ is noisy, but actual HAZ is also very noisy because many DHS respondents in LMICs don't know their child's birthday or even month of birth, as shown in a previous paper we now reference. We have added references to this problem in the discussion section:

“DHS anthropometric indicators are known to be measured with error, with problems in measuring height, weight and even child age accurately,^{49,50} which may reduce the precision of our estimates.”

Why not use weight-for-age as a measure of acute malnutrition?

This indicator is no longer preferred by nutritionists because it mixes up chronic nutrition (HAZ) and acute nutrition (WHZ). It's dynamic linkages to shocks are therefore unclear. Moreover, as we stated above, child age is actually also measured with considerable error in the DHS, as shown by the study below by Larsen et al.. So using weight for age won't help with measurement error in the dependent variables – it's not a superior measure in the low error sense, and also confounds acute and chronic undernutrition, so we elect not to use it here.

Larsen, A. F., Headey, D., & Masters, W. A. (2019). Misreporting Month of Birth: Diagnosis and Implications for Research on Nutrition and Early Childhood in Developing Countries. *Demography*, 56, 707-728.

Reviewer comments, second round

Reviewer #1 (Remarks to the Author):

I appreciate the authors' revisions to the manuscript, including their responses to my and other reviewer comments. In my view, this paper makes important contributions to the literatures on undernutrition in LMIC contexts and on the health and economic effects of macro shocks. I look forward to citing it.

Brian

Reviewer #2 (Remarks to the Author):

No further suggestions.

Reviewer #3 (Remarks to the Author):

Revision is fine